# Clinical Applicability of a Textile 1-Lead ECG Device for Overnight Monitoring

**DOI:** 10.3390/s19112436

**Published:** 2019-05-28

**Authors:** Piero Fontana, Neusa R. Adão Martins, Martin Camenzind, René M. Rossi, Florent Baty, Maximilian Boesch, Otto D. Schoch, Martin H. Brutsche, Simon Annaheim

**Affiliations:** 1Empa, Laboratory for Biomimetic Membranes and Textiles, Lerchenfeldstrasse 5, 9014 St. Gallen, Switzerland; piero.fontana@empa.ch (P.F.); neusa.martins@empa.ch (N.R.A.M.); martin.camenzind@empa.ch (M.C.); rene.rossi@empa.ch (R.M.R.); 2High-performance Scientific GmbH, Wilenstrasse 24, 8832 Wilen bei Wollerau, Switzerland; 3Campo Grande, Faculdade de Ciências, Universidade de Lisboa, 1749-016 Lisboa, Portugal; 4Kantonsspital St. Gallen, Klinik für Pneumologie/Schlafmedizin, Rorschacher Strasse 95, 9007 St. Gallen, Switzerland; florent.baty@kssg.ch (F.B.); maximilian.boesch@kssg.ch (M.B.); otto.schoch@kssg.ch (O.D.S.); martin.brutsche@kssg.ch (M.H.B.)

**Keywords:** validation, long-term electrocardiogram, textile electrodes, ECG-belt, polysomnography, clinical applicability

## Abstract

Even for 1-lead electrocardiography (ECG), single-use gel conductive electrodes are employed in a clinical setting. However, gel electrodes show limited applicability for long-term monitoring due to skin irritation and detachment. In the present study, we investigated the validity of a textile ECG-belt suitable for long-term measurements in clinical use. In order to assess the signal quality and validity of the ECG-belt during sleep, 242 patients (186 males and 56 females, age 52 (interquartile range 42–60) years, body mass index 29 (interquartile range 26–33) kg·m^−2^) with suspected sleep apnoea underwent overnight polysomnography including standard 1-lead ECG recording. The single intervals between R-peaks (RR-intervals) were calculated from the ECG-signals. We found a mean difference for average RR-intervals of −2.9 ms, a standard error of estimate of 0.39%, as well as a Pearson *r* of 0.91. Furthermore, we found that the validity of the ECG-belt decreases when lying on the side, which was potentially due to the fitting of the belt. In conclusion, the validity of RR-interval measurements using the ECG-belt is high and it may be further improved for future applications by optimizing wear fitting.

## 1. Introduction

Electrocardiography (ECG) is widely used for evaluating a variety of cardio-circulatory and other health conditions. 1-lead ECG signals provide relevant information for R-peak detection, calculation of RR-intervals and evaluation of heart rate variability (HRV). The gold standard electrodes for ECG signal acquisition are single-use gel conductive electrodes consisting of Ag/AgCl, enclosed by a conductive gel. The gel is surrounded by biocompatible glue, which is harnessed to fix the electrodes on the skin. While gel conductive electrodes are cheap and easy to use, their application is limited to short-term monitoring due to progressive drying. Furthermore, their use is restricted to stationary monitoring and requires trained health care personnel. Consequently, other systems employing different kinds of measurement techniques, such as patch electrodes, capacitive electrodes, or photoplethysmography have been developed and commercialized [1,2,3,4]. Other than gel electrodes, these systems are re-usable, offer a high level of flexibility and provide better wearing comfort. Furthermore, they allow long-term measurements and reduce the risk of skin irritations. However, their use is potentially hampered by movement artefacts (not glued to the skin) and cleaning issues (e.g., mechanical stability after multiple washing cycles).

Recently, Weder et al. [5] reported the development of a portable ECG measuring device (ECG-belt), contemplated to pass the disadvantages of current wearable systems. The ECG-belt employs embroidered, self-humidifying electrodes with Ag/Ti coating for long-term ECG-monitoring. It meets all the requirements concerning cytotoxicity and signal stability. Furthermore, it offers high practicability, such as the possibility of reutilization, long-time durability, washing fastness, durability, and easy application. Using the ECG-belt, a 1-lead ECG can be recorded, offering long-term monitoring of RR-intervals including the calculation of derived parameters, such as heart rate and HRV. Therefore, the ECG-belt is of particular interest for clinical setups involving the need for long-term cardiac monitoring based on RR-intervals.

A relevant clinical setup of interest for long-term ECG recordings is sleep apnoea diagnosis, which is characterized by impaired breathing, accompanied by enhanced sympathetic tone [6,7,8,9,10,11] and changes in HRV [12]. In particular, periodic oscillations of the heart rate, systolic blood pressure, and alveolar ventilation may be common incidents in patients with sleep apnoea syndromes and may vary significantly during sleep [13]. For sleep apnoea diagnosis, overnight laboratory-based polysomnography (PSG) involving an ECG based on gel conductive electrodes is the unrivalled standard-of-care. However, PSG is elaborate and costly, and involves a number of stationary measurement devices, making it technically demanding [14]. Therefore, easier solutions for screening and patient follow-up purposes, such as ECG measurements using the ECG-belt are warranted [2,15]. To date, it remains unclear whether wearing the ECG-belt in clinical set up for overnight measurements is feasible and whether the ECG-belt delivers valid ECG recordings when compared to conventional gel electrodes used during PSG. Therefore, we sought to compare RR-interval measurements obtained during PSG and the ECG-belt during overnight PSG. Furthermore, we analyzed the occurrence of artefacts and assessed the influence of body position on the incidence of artefacts. We hypothesized that using the ECG-belt is feasible in clinical practice and that the ECG-belt delivers valid measures of autonomic cardiac regulation when compared to conventional gel electrodes.

## 2. Materials and Methods

### 2.1. Study Design and Overview

In order to validate and test the feasibility of the ECG-belt in a clinical setup, 242 patients with suspected sleep apnoea were included in the study (Table 1). Before an overnight stay, patients were advised to take their respective standard medication, before they underwent PSG according to the recommendations of American College of Physicians [16]. This included long-term ECG-monitoring using conventional gel electrodes and concomitant ECG-monitoring with the ECG-belt (Figure 1). ECG-data acquired using conventional gel electrodes (PSG) and the embroidered electrodes (ECG-belt) were compared. Average overnight measurement duration was 6.0 h (SD 0.8). As the primary outcome, we calculated measures of validity based on mean RR-intervals (RRmean; ms). The study was performed in strict accordance with the standards laid down in the 1964 Declaration of Helsinki including amendments (newest version 2013), the principles of Good Clinical Practice, as well as the Swiss legal requirements. Approval of the examination protocol was obtained from the ethics committee of Kanton St. Gallen (EKSG Nr.15/140).

### 2.2. Study Participants

We recruited a total of 245 symptomatic patients with suspected sleep apnoea assigned for PSG at the sleep center of the Kantonsspital St. Gallen, Switzerland. Three patients did not complete the study and were excluded. Characteristics of the study participants and the results of PSG are listed in Table 1. Body position analysis was performed in a subgroup of 7 females and 9 males. Written informed consent was obtained from all patients following a full explanation of the purpose and nature of the study and the potential risks and discomforts associated with participation.

### 2.3. PSG and Conventional ECG (Gel Electrodes, Gold Standard)

For PSG, a standard polysomnography system including snap on cables (Embla N7000, Embla Systems Inc., Broomfield, CO, USA) and gel electrodes (Ambu BlueSensor N, Ambu GmbH, Bad Nauheim, Germany) was used. Placement of electrodes is depicted in Figure 1.

### 2.4. ECG-Belt (Embroidered Electrodes)

The ECG-belt used in this study was previously described by Weder et al. [5]. Placement of the ECG-belt and the gel electrodes are shown in Figure 1. The ECG-belt consists of a semi-elastic polyester belt (Unico Swiss Tex GmbH, Alpnachstad, Switzerland) with directly embroidered Ag/Ti-coated PET yarn (Serge Ferrari Tersuisse AG, Emmenbrücke, Switzerland), forming the electrodes. These electrodes exhibit a low electrical resistance of typically <1 kΩ, provide good skin contact (textile belt and flexible, embroidered electrodes based on PET fibers coated by approx. 100 nm Ag and approx. 4–7 nm Titanium), are supposed to overcome skin irritations associated with gluing, and due to moistening, enable good signal quality also on dry skin and at rest. The two 20 × 70 mm electrodes are incorporated into a textile “wearable” belt with stretchable parts and trimmed to adjust the size. The wearable includes a wetting system (small water tank incorporated in the area between the electrodes connected to pads matching the size of the electrodes with a high water vapor permeable membrane towards the electrodes), which delivers approximately 1 to 2 g of water per day.

The stored water evaporates mainly due to the body heat of the wearer, increases the electrical conductivity and thus, reduces motion-related artifacts. It allows for continuous measurements over five to seven days. Signals acquired by the ECG-belt are stored on an attached logging device and can be simultaneously transmitted to an external device depending on the capability of the logger. Signals were acquired with a sampling rate of 200 Hz and signal quality was evaluated afterwards. Data was acquired by the PSG system. The combination of Ag and Ti in a low-humidity environment was found to yield a good ECG signal [5]. All materials used for the belt were skin-friendly and non-cytotoxic. In the present study, 16 different ECG-belts (same build) were used. For each belt, measurement quality was continuously evaluated after each use. Seven belts were excluded from the study due to reduced data quality or obvious damages following extensive use.

### 2.5. Data Processing and Statistical Analysis

The sample size was calculated based on the precision needed to assess the confidence interval of the 95% limits of agreement between the ECG-belt and the gold standard gel electrodes. For every patient, we recorded a 1-lead ECG using both methods of measurement in parallel. The quality of the ECG signal was mainly affected by motion artefacts, baseline wander and high-frequency noise. Therefore, second-order Butterworth filters (low-pass and high-pass) were applied to estimate low and high-frequency noise. Based on this data, the signal to noise ratio was calculated for low (SNRlf) and high-frequency noise (SNRhf). The band-pass filtered signal was taken as the noise-free reference signal. Baseline wander (BLW) was calculated based on the amplitude of the sinusoidal approximation of the baseline fluctuations. The effect of non-Gaussian noise (such as movement and body posture related artefacts) is indicated by the artefacts of R-peak detection as explained below.

RR-intervals were detected from the raw ECG signal using Kubios HRV Premium Software, Version 2.2 (Kubios Oy, Kuopios, Finland). The software includes a detection algorithm for the QRS complex of the ECG signal consisting of a preprocessing part followed by the decision rules. The preprocessing part includes bandpass filtering of the ECG signal to reduce power line noise, baseline wanders and other noise components, squaring of the data samples to highlight peaks and moving average filtering to smooth close-by peaks. The decision rules include amplitude threshold and comparison to an expected value between adjacent R-waves. Both of these rules are adjusted adaptively every time a new R-wave is acceptably detected. Before R-wave time instant extraction, the R-wave is interpolated at 2000 Hz to improve the time resolution of the detection. This up-sampling will significantly improve the time resolution of R-wave detection when the sampling rate of the ECG is relatively low (Kubios HRV Premium Software manual, Version 3.2, https://www.kubios.com/downloads/Kubios_HRV_Users_Guide.pdf). We then applied a filter on the RR-interval data set according to the Task Force of the European Society of Cardiology and the North American Society of Pacing and Electrophysiology [17]. RR-intervals shorter than 300 and longer than 1500 ms were excluded from the analysis as well as RR-intervals differing more than 20% from the median of the preceding, or the following ten RR-intervals. We then calculated mean RR-intervals (RRmean) as the average of the overnight measurement period. Average data of all subjects were checked for normality and homoscedasticity using Kolmogorov-Smirnov and Levene’s statistics, respectively. We compared the data obtained from the two measurement methods using Student’s paired *t*-test. We calculated mean differences between the ECG-belt and the gold standard, root mean square, standard error of measurement, and Pearson’s *r*. Moreover, Bland-Altman plots were generated, and linear regression analysis was performed. Outliers were defined as values differing more than the 95% confidence interval (111 ms, −117 ms) from Bland-Altman mean difference. Statistical significance was set to *p* < 0.05. Poincaré analysis was applied to the filtered RR-intervals, and the standard descriptors SD1 and SD2 were computed according to the formulae published by Piskorski and Guzik [18]. For all body positions, a tolerance of 45° was used [19]. All statistical analyses were performed using SPSS 17.0 for Windows (SPSS Inc., Chicago, IL, USA). Filters were programmed using MATLAB R2013b (The MathWorks, Inc., Natick, MA, USA). Data are presented as mean value ± standard deviation.

## 3. Results

In the present study, we report the data of 242 study participants. In Figure 2, a representative example of ECG waveforms for gel electrodes and the ECG-belt is shown. Data for signal quality is presented in Table 2. A good agreement was found for mean RR-intervals recorded by the two measurement methods (Table 3; Appendix A). When the outliers (six in total, see Figure 3 and Figure 4) were excluded from the analysis, the difference between the two measurements became statistically significant (*p* < 0.001). However, the absolute difference changed only slightly from 3 ms to 4 ms, which is not considered to be a relevant difference with regard to HRV analysis. The outliers occurred due to a bad connection in one case (no detection of any ECG signals) and due to multiple uses of the ECG belts in the other five cases. In all of the five cases, the ECG belt had been used at least 17 times or more. Furthermore, no difference could be detected between the two measurement methods upon stratification for sex (male: 952.78 ± 135.83 ms vs. 951.45 ± 143.40 ms *(p* = 0.778), female: 917.38 ± 120.74 vs. 906.54 ± 140.65 ms (*p* = 0.317) for gel electrodes and the ECG-belt, respectively).

Linear regression analysis and Bland-Altman plots are shown in Figure 3 and Figure 4. Coefficient of determination for mean RR-intervals was 0.77 (linear regression), whereas 95% confidence interval for Bland-Altman was −2.94 ± 113.93 ms. Outliers with negative y-values apparent in Figure 4 turned out to have high artefact percentages for the ECG-belt (41.7 to 66.6%) and low artefact percentages for the gel electrodes (0.2 to 1.1%).

In Figure 5, quality measures for individual ECG-belts are depicted. The median artefact was below 5% for 13 out of 16 belts including belts that have been used more than 20 times (i.e., ECG-belt 16–18). The difference in artefacts between ECG-belts and gel electrodes emphasizes the artefacts associated with the ECG-belt. Finally, the average change in artefacts gives an indication of the change in artefacts to be expected with multiple usages of the ECG-belt. A low standard deviation (i.e., lower than 10%) indicates a consistent change in artefacts while a high standard deviation stands for a high variation in the change of artefacts. The results show that the ECG-belts can be used up to 25 times without big changes in artefacts (i.e., ECG-belt 18) while the ECG-belt can be worn out even at a lower number of usage if not handled carefully (i.e., ECG-belt 26). Nevertheless, it can be inferred from the data that a usage of up to 13 times results in a median artefact of 5.4% while for the more extensive belt use (number of use >13), the median for artefacts increased to 27.8%. The comparison of Poincaré metrics, SD1 and SD2, obtained based on RR-intervals measured using from the ECG-belt and gel electrodes revealed a Pearson *r* of 0.46 (*p* < 0.001) for SD1 and 0.45 (*p* < 0.001) for SD2 including the whole dataset (n = 242; Appendix A). For belt uses <14 (n = 180), a Pearson *r* of 0.68 (*p* < 0.001) and 0.79 (*p* < 0.001) was found for SD1 and SD2, respectively.

Aside from the number of uses of the ECG-belt, body position had an influence on the occurrence of artefacts, also. The influence of body position (subgroup analysis) on the percentage of artefacts is shown in Figure 6 (Appendix A).

## 4. Discussion

In the present study, we assessed signal quality and the validity of RR-interval measurements of a textile single lead ECG-device with embroidered electrodes (ECG-belt, [5]) during PSG-recordings in 242 patients with suspected sleep apnoea (Figure 1). While the technical details of the ECG-belt have been reported earlier [5], this is the first report describing the use of the ECG-belt in a clinical setup. For determining the validity of RR-interval measurements, we compared RR-interval measurements of conventional gel electrodes and the ECG-belt during an overnight stay in a hospital (‘side-by-side’ analysis). Our main findings indicate that the validity of RR-interval measurements using the ECG-belt is high and the ECG-belt is a promising device for monitoring heart rate variability in a clinical environment. Furthermore, our findings indicate that the measurement quality of the ECG-belt may be influenced by body position of the patient and that the maximal number of uses of a belt is limited and may depend on the handling.

Signal quality was assessed by means of high- and low-frequency SNR and baseline wander (Table 2). Particularly, the low-frequency SNR and baseline wander are critical parameters to be considered as they affect R-peak detection. Even though statistically significant differences were detected between signal quality for gel electrodes and ECG-belt, no significant effect was found for R-peak detection and the respective calculation of RR-intervals (Table 3). Furthermore, evidence for the high validity of RR-interval measurements using the ECG-belt comes from our observation that mean values of the ECG-belt measurements only differed by 0.39% from respective reference values (gel electrodes) and that the standard deviations were similar between the two methods (Table 3). In addition, high validity is indicated by a low root mean square as well as a low standard error of the estimate. Furthermore, Pearson *r* for comparisons between the ECG-belt and the gold standard was in a similar range as for comparisons of other wearable devices and gel electrodes ([20]; Table 3). Corroborating these data, regression analysis with a high coefficient of determination showed high validity of the RR-interval measurements using the ECG-belt (Figure 3). Finally, the high validity of the measurements using the ECG-belt is confirmed by the Bland-Altman plot with narrow 95% confidence intervals (upper CI 111 ms, lower CI −117 ms). In particular, Bland-Altman analysis indicates a small fluctuation range of differences as well as no trend to a systematic over- or underestimation of values assessed by the ECG-belt. However, in our Bland-Altman plot, six outliers were apparent, all of them revealed large differences in artefact values between the ECG-belt and the gel electrodes. Our findings are in line with the results of Georgiou et al. [20], who reported in his meta-analysis several studies with similar correlation coefficients for ECG measurements using gel electrodes and portable ECG devices. In addition, high values in SNR for high frequency for both systems confirmed high signal quality, even though a statistically significant difference was observed between gel electrodes and ECG-belt.

In relation to the findings of Akintola et al. [21], who observed a mean artefact percentage of 19% for a portable ECG measurement, our artefact percentages were rather low. However, a general trend to an increase in artefacts was observed after 13 applications. Strikingly, Pearson *r* increased for comparisons between measurement methods for Poincaré SD1 and SD2 when the dataset was condensed to data originating from belts used less than 14 times. Thus, the number of uses seems to be a strong determinant of data quality. Nevertheless, for some ECG-belts, acceptable signal quality and low artefacts were found even after 20 applications and more. On the other hand, some measurements with high artefacts were observed for ECG-belts with a number of use lower than 13. These cases may be explainable by two facts. Some of the cases are specific for the patient since patient specific maximal artefact values for gel electrodes were similarly high. In other cases, where the maximal number of artefacts for the ECG-belt was markedly higher than for gel electrodes, we hypothesize that this is due to the suboptimal fitting of the ECG-belt. It is also important to note that the ECG-belts were not always used in a similar extent. While 5 belts were used up to 19 times, only three belts were used more than that (Figure 5). Therefore, for 20 applications or more, the number of belts is relatively small and, thus, data may be affected by the small sample size. The increased number of artefacts for 14 and more applications of a belt is likely to result from the durability of the material and components themselves or the handling. Thus, proper handling standards should be established. The quality of the ECG-belt data is also influenced by the body position of the patients. Of note, the occurrence of artefacts was similar for gel electrodes and the ECG-belt for a supine and prone body position. Therefore, our data clearly shows that the measurement technology itself (e.g., electrodes and electronics) is as valid as the gold standard. However, for patients lying either on the left or on the right body side, the percentage of artefacts was higher than for the prone and supine body position for the ECG-belt, whereas it was similar for the gel electrodes. We hypothesize that lying on the body side leads to a disruption of the contact between the electrodes and the skin through mechanical deformation of the belt. Consequently, further improvements of the ECG-belt should focus on improving the fitting of the textile belt or the shape of the electrodes in order to avoid a disruption of the contact of the electrodes and the thorax. A better fitting may be achieved, for instance, by choosing a more flexible material for the textile belt and by adjusting the shape of the belt.

### Clinical Relevance

As shown in previous work [22], HRV indices derived from ECG signals can constitute clinically valuable tools for detecting differences among varying levels of severity in sleep apnoea patients. Whole night time-domain and frequency-domain analyses can be used to monitor the HRV nocturnal evolution. These methods generate a set of indices which can be combined into multivariate scores for the prediction of apnoea severity. Besides sleep apnoea detection, future clinical applications of the ECG-belt may include long term RR-interval measurements in cardiology as well as autonomic nervous system assessments (e.g., for evaluating the recovery status) in Sports Medicine. Nevertheless, this study evaluated the technical validation of the ECG-belt which included the assessment of the signal quality as well as the comparison of RR-intervals calculated based on the ECG signals obtained from both systems. The further evaluation of HRV indices will be investigated in future studies.

## 5. Conclusions

Based on our findings, we conclude that the ECG-belt provides RR-interval measurements of high validity and that the device can be incorporated into future clinical studies. In particular, HRV-data assessed by the ECG-belt may be used to identify the severity of the obstructive sleep apnoea syndrome. Furthermore, we conclude that measurement quality can be further improved by incorporating a more flexible material into the textile belt and optionally, by improving the shape of the textile. Finally, we conclude that handling of the belt should be systematically evaluated in order to increase the longevity of the material and, therefore, extend the lifespan of the ECG-belt.

## Figures and Tables

**Figure 1 sensors-19-02436-f001:**
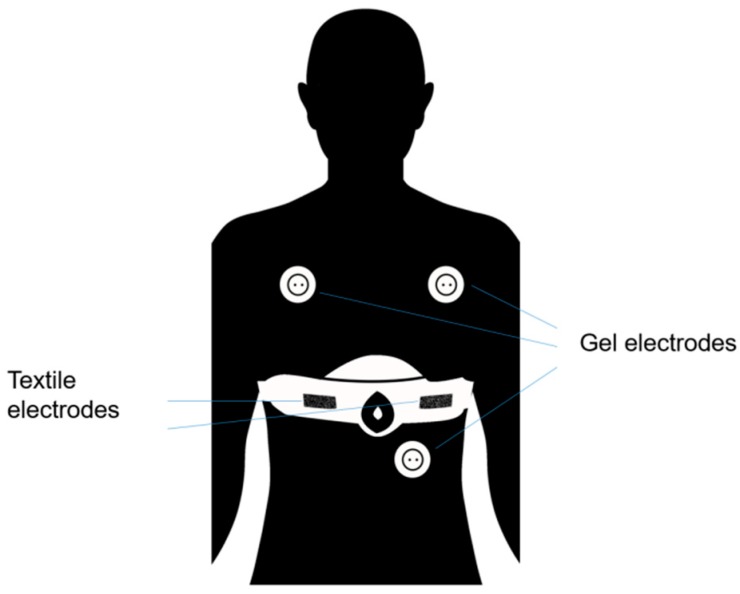
Placement of the gel electrodes and the ECG-belt.

**Figure 2 sensors-19-02436-f002:**
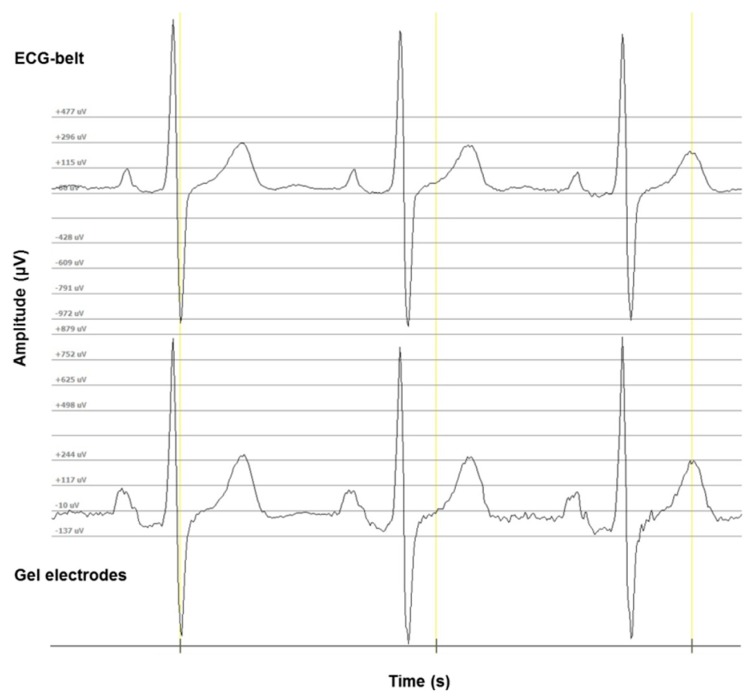
A representative sample of ECG waveforms for gel electrodes and the ECG-belt.

**Figure 3 sensors-19-02436-f003:**
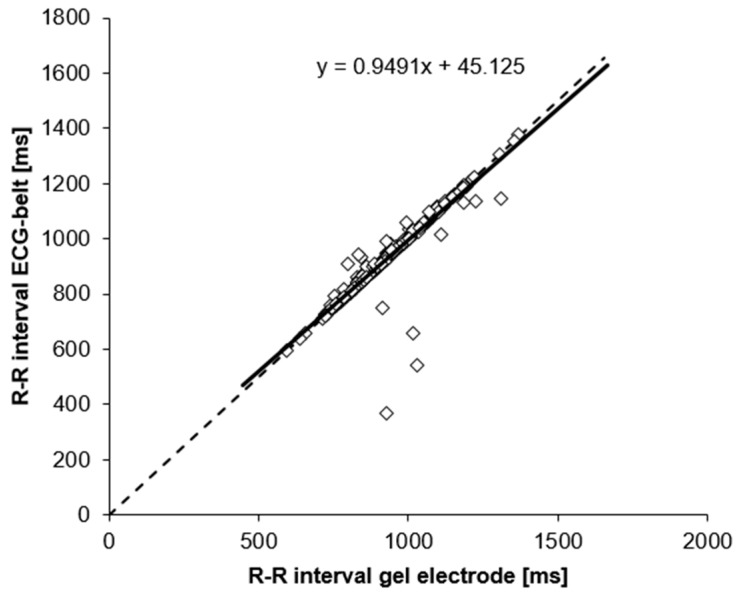
Line of identity (dashed line) for RR-intervals measured using gel electrodes and the ECG-belt. Solid line, regression line incl. fit equation.

**Figure 4 sensors-19-02436-f004:**
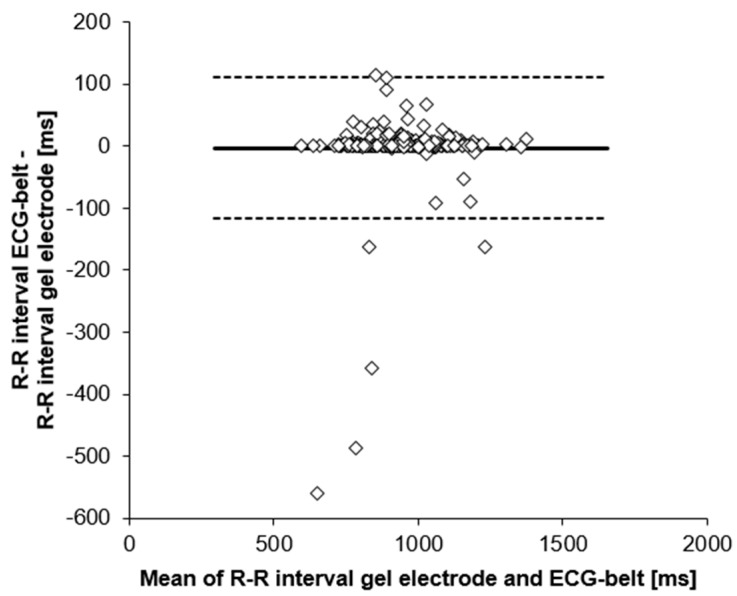
Bland-Altman plot of RR-intervals measured using gel electrodes and the ECG-belt. Solid line, mean difference; dashed lines, upper and lower 95% confidence intervals.

**Figure 5 sensors-19-02436-f005:**
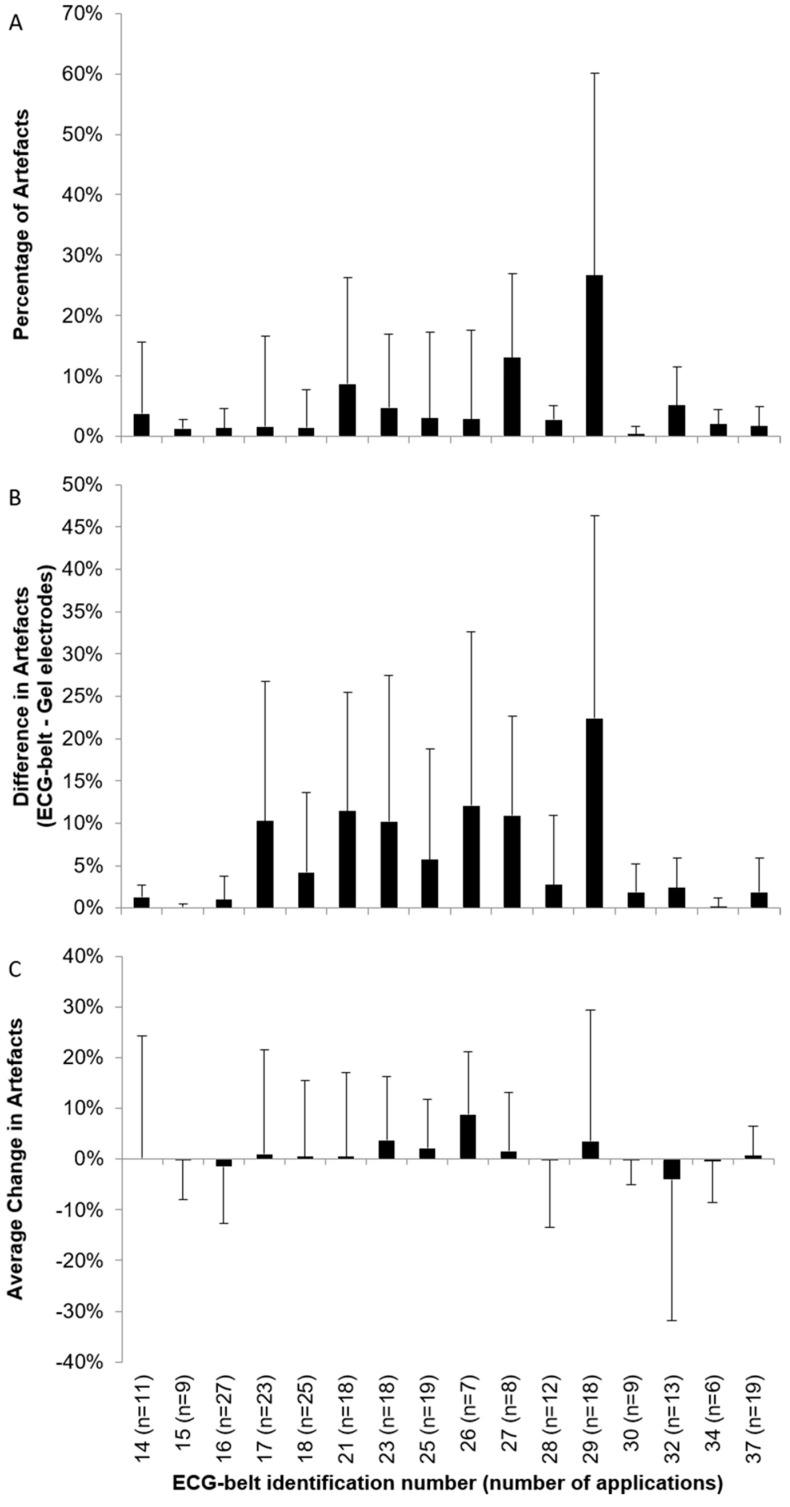
Data about artefacts observed for individual ECG-belt and number of use (n). (**A**) The median and interquartile range of artefacts observed for individual ECG-belts. (**B**) Average difference (standard deviation) in artefacts between ECG-belt and gel electrodes and standard deviation to emphasize ECG-belt associated artefacts. (**C**) The average change in artefacts (standard deviation) for consecutive applications of individual ECG-belts.

**Figure 6 sensors-19-02436-f006:**
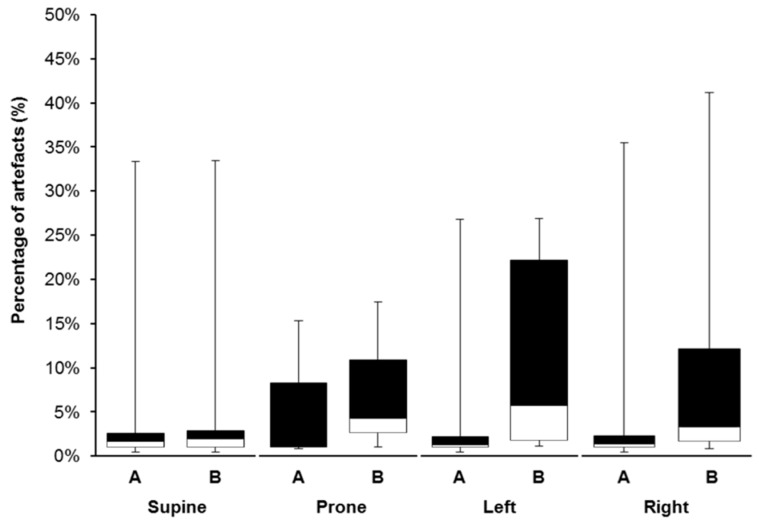
Influence of body position on the occurrence of artefacts for RR-interval measurements using gel electrodes and the ECG-belt. The influence of body position (supine, prone, left body side, right body side) on measurement quality is presented. Data relate to 13 or fewer applications of a belt. A, gel electrodes; B, ECG-belt. Supine, patient lying on the back; prone, patient lying on the chest; left and right, patient lying laterally Black part of boxes indicates upper quartile; white part of boxes indicates the lower quartile. Whiskers represent data for maximal values.

**Table 1 sensors-19-02436-t001:** Baseline characteristics of patients with suspected sleep apnoea.

n	242	
Age (y)	52 [43–61]	
male / female	186/56	
BMI (kg·m^−2^)	29 [25.5–32.5]	
ESS	9 [6–12]	
ODI (h^−1^)	17 [0.9–33.2]	
AHI (h^−1^)	21 [4.4–37.6]	
Type of apnoea	Obstructive	156
	Central	5
	Mixed	35
	Unspecified	1
	No apnoea detected	44

Data represent median [interquartile range]; BMI, body mass index; ESS, Epworth sleepiness scale; ODI, oxygen desaturation index; AHI, apnoea-hyperpnoea index. The information about the type of apnoea is missing for one patient.

**Table 2 sensors-19-02436-t002:** Mean values for a signal to noise ratio for low (SNRlf) and high-frequency noise (SNRhf), as well as baseline wander (BLW).

		Gel Electrodes	ECG-Belt	*p*
**SNRhf**	**(dB)**	21	17	<0.001
**SD**	**(dB)**	3	6	
**SNRlf**	**(dB)**	12	0	<0.001
**SD**	**(dB)**	5	5	
**BLW**	**(mV)**	0.03	0.30	<0.001
**SD**	**(mV)**	0.02	0.43	

**Table 3 sensors-19-02436-t003:** Mean values of filtered averaged RR-intervals for measurements using a gel electrode and measurements using the ECG-belt.

		Mean			*p*	RMS	SEE	Pearson *r*
		A	B	Diff				
**RR_mean_**	**(ms)**	944.84	941.90	−2.94	0.429	57.79	0.39%	0.91
**SD**	**(ms)**	133.20	138.87	57.84				

A: measurements using a gel electrode; B: measurements using the ECG-belt; RRmean, filtered aver-aged RR-intervals; SD, standard deviation; Diff, mean difference; *p*, *p*-value; RMS, root mean square; SEE, standard error of estimate.

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
