# Peer review of "Clinical Applicability of a Textile 1-Lead ECG Device for Overnight Monitoring"

_sensors, 2019, doi:10.3390/s19112436_

Reviewer 1 Report

The paper is, generally, well described and clear enough.

Moreover, it requires more details on some points/aspects which have not been dealt with or have not sufficiently covered in the study. In particular:

1.      For the diagnosis of cardiovascular diseases, the characteristics of the ECG signal (in terms of QRS complex, P-wave, T-wave, etc) are important. Are the ECGs acquired with these electrodes qualitatively good for diagnosis?

2.      Are the electrodes able to acquire the typical characteristics and features of ECG signal with high SNR ratio in different states of motion?

3.      What is the SNR ratio?

4.      In what range the baseline fluctuations occur?

5.      Is the signal quality time dependent? A study of the ECG signal day-to-day should be advisable

6.      Does the electrode combine the potential for use in log-term wearable dynamic cardiac activity monitoring systems with the convenience for use in home health care in the case of high-risk people?

7.      Adopting these electrodes, a computer aided detection (CAD) system or a physician are able to detect sudden onset of heart diseases?

8.      From the study it isn’t clear the potential usefulness of the proposed electrodes for routine electrophysiological activity monitoring of heart and of other vital organs.

9. For a full use of electrodes not only the resting-state ECG signal should be analyzed but also the motion artifacts that result from walking and swinging of the subject’s arms should be investigated

Author Response

Thank you very much for your valuable inputs. Please see document for point-by-point responses attached.

Reviewer 2 Report

The paper presents well organized research in the area which is interesting for any long-term electrophysiological diagnoses. The presentation style is good, however needs to be corrected to provide the readers wit most accurate information of what was done or recommendation for future use. In my opinion, starting from the title, the paper is 'overpromissing' and will be a fair report of your research after few corrections. Main concerns are:

line 2: the title is promising much more than has been proven. In fact only RR interval-based diagnostics has been tested and compared, while other e.g. contour or ST-segment analyses are not addressed.

line 71: Usually the assessment of autonomic cardiac regulation in sleep is based on HRV taking into account only sinus rhythm. Arrhythmia of any type should be excluded as non-sinus beats are out of control of ANS. Please describe whether the loss performance in morphology recognition with the belt comparing to the conventional ECG is small enough to correctly identify extra-sinus beats and exclude their RR from calculations.
line 136: Depending on the preceding content of the ECG, the beats may be detected either prematurely (e.g. if their amplitude increases) or with a few millisecond delay. It is then interesting whether the Kubios HRV Premium Software used for measurements of RR intervals does some corrections (e.g. parabola fitting etc.). 

line 185-6: Fig. 5 digits along the horizontal axis are unclear. were belts no. 1-5 used 16 times and no. 19 - 5 times? More important than detailed statistics (such as (P < 0.001) etc.) is the discussion which tells the opposite to the plot.
On the other hand it would be interesting why the electrodes wear out. Did you check their structure with a microscope or performed other parameter estimation after the >20-th use?
line 185: The vertical axis label is also unclear. E.g.: were there 68% of artefacts with belt no. 3 after 16-th application?

line 208: so what is new with regard to [5]?
- details on the belt has been reported there (?)
- a 'device': as I understood you used the same commercial recording device and ECG interpretation software,
It would be then more accurate to report on 'testing of the belt (described in [5]) in a comparative clinical setup for RR interval based sleep apnoea monitoring'.  
line 234: Before giving a recommendation to use of the belt instead of gel electrodes, the discussion needs extension beyond the pure statistics of RR errors. Please identify and explain the outliers. Please identify arrhythmia events (in 243 patients x 6 hours you should have some) and compare the behavior of the RR analysis. Sleep apnoea conditions may trigger arrhythmias due to hypoxia.

Author Response

Thank you very much for your valuable inputs. Please see document for point-by-point responses attached.

Round  2

Reviewer 1 Report

Authors only replied to very few of my arguments.

The objections highlighted in my previous report are very important to assess the ECG device validity in the routine activity monitoring of heart. In the clinical routine, the adoption of ECG devices which enable the evaluation of only a few parameters, not ensuring a signal quality which is appropriate for the monitoring of patient heart condition, is unimaginable.

The use of 1 lead ECG signal should not be a limitation for the analysis of signal quality in fact, most of the computer aided detection and diagnosis systems, which are indicated in literature, adopt 1 lead ECG signal.

Even if the ECG signal acquisition was done during sleep (as authors said), artifacts (patients aren’t motionless/stationary for all the night), baseline fluctuations and other type of noises should be considered.

For the above reasons, authors should resolve all the criticalities indicated in my previous report.  

Moreover, to assess the obtained values (R point localizations, RR interval, etc) in the international panorama, the sensitivity, the specificity, the detection error rate and the FROC curve should be evaluated.

Author Response

Please find the point-by-point response in the document attached.

Reviewer 2 Report

Thank you for improving the manuscript accordingly to my suggestions.

Minor technical issues could be corrected at the editorial stage without the need for further review such as:

'...due to bad connection in one case [...] and due to [...] in the other six cases. In all the FIVE (?) cases,
'Nevertheless, for some ECG-beltS (?), acceptable...'

Otherwise, the paper is now ready to publish.

Author Response

Please find the point-by-point response in the document attached.

Round  3

Reviewer 1 Report

Authors have to revise tab.1 (so to place it in a single page) and fig5 (labels and caption)